# Alphaviruses in Immunotherapy and Anticancer Therapy

**DOI:** 10.3390/biomedicines10092263

**Published:** 2022-09-13

**Authors:** Kenneth Lundstrom

**Affiliations:** Pan Therapeutics, 1095 Lutry, Switzerland; lundstromkenneth@gmail.com

**Keywords:** alphavirus vectors, recombinant particles, RNA delivery, DNA replicons, cancer immunotherapy, therapeutic efficacy, tumor protection, clinical trials

## Abstract

Alphaviruses have been engineered as expression vectors for vaccine development and gene therapy. Due to the feature of RNA self-replication, alphaviruses can provide exceptional direct cytoplasmic expression of transgenes based on the delivery of recombinant particles, naked or nanoparticle-encapsulated RNA or plasmid-based DNA replicons. Alphavirus vectors have been utilized for the expression of various antigens targeting different types of cancers, and cytotoxic and antitumor genes. The most common alphavirus vectors are based on the Semliki Forest virus, Sindbis virus and Venezuelan equine encephalitis virus, but the oncolytic M1 alphavirus has also been used. Delivery of immunostimulatory cytokine genes has been the basis for immunotherapy demonstrating efficacy in different animal tumor models for brain, breast, cervical, colon, lung, ovarian, pancreatic, prostate and skin cancers. Typically, therapeutic effects including tumor regression, tumor eradication and complete cure as well as protection against tumor challenges have been observed. Alphavirus vectors have also been subjected to clinical evaluations. For example, therapeutic responses in all cervical cancer patients treated with an alphavirus vector expressing the human papilloma virus E6 and E7 envelope proteins have been achieved.

## 1. Introduction

During the last decade, immunotherapy has become an attractive alternative for cancer therapy [1]. In this context, viral vectors have also proven useful for immunotherapeutic applications [2]. Alphaviruses have frequently been engineered for the overexpression of suitable antigens and immunostimulatory genes for vaccine development and cancer therapy [3]. Additionally, the expression of cytotoxic and antitumor genes has been used for cancer therapy applications. Semliki Forest virus (SFV) [4], Sindbis virus (SIN) [5] and Venezuelan equine encephalitis virus (VEE) [6] are most commonly used for the engineering of expression systems. Additionally, the naturally occurring oncolytic alphavirus M1 [7] and engineered versions based on SFV and SIN vectors [8] have been utilized for cancer therapy. The evaluation of efficacy in appropriate animal models has provided proof of concept before conducting clinical trials.

In this review, the alphavirus lifecycle and different expression vector systems are described. Examples are also presented from preclinical studies in animal models for various types of cancers. A summary of the clinical evaluation of safety and efficacy of prophylactic and therapeutic efficacy are also included.

## 2. Alphavirus Lifecycle and Expression Vector Systems

Alphaviruses possess an enveloped structure of capsid and spike proteins encapsulating a single-stranded RNA (ssRNA) genome of positive polarity [9]. Upon the infection of host cells, the alphavirus ssRNA is released into the cytoplasm, where translation can immediately occur requiring no delivery of RNA to the nucleus as is the case for other RNA viruses such as the influenza virus and DNA viruses (Figure 1). In the cytoplasm, efficient self-replication occurs through a minus-strand RNA template leading to the accumulation of approximately 10^6^ copies of subgenomic RNA per cell. Together with the utilization of the highly efficient 26S subgenomic promoter, high-level expression of viral proteins occurs [10]. The RNA self-replication and high-level expression of alphavirus structural proteins generate high-titer virus progeny. Nucleocapsids comprising the capsid protein harboring full-length alphavirus RNA are transported to the cell surface, where the envelope proteins are attached, and mature viral particles are released by budding.

In the case of expression systems, the focus is on the expression of heterologous genes (Figure 2). In the context of replication-deficient alphavirus particles, the structural protein genes have been replaced by the gene of interest (GoI), and a helper vector is engaged in providing the structural proteins in trans (Figure 2A). Cotransfection of in vitro transcribed RNA from expression and helper vectors into baby hamster kidney (BHK) cells leads to the production of recombinant particles. As the RNA packing signal is located in one of the genes coding for the nonstructural proteins (nsPs) in the nsP2 gene of SFV and nsP1 of SIN [11], uniquely, RNA from the expression vector is packaged into viral particles, providing expression of the GoI but not the structural protein genes and thereby, eliminating any production of viral progeny. In contrast, introduction of a second 26S subgenomic promoter and the GoI into the full-length alphavirus RNA genome, either downstream of the nsP or the structural protein genes, generates replication-competent particles capable of both high-level GoI expression and viral progeny production (Figure 2B). In addition to the application of recombinant particles, RNA replicons can also be used for GoI expression. As has been demonstrated for the recent BNT162b2 [12] and mRNA-1273 [13] COVID-19 vaccines, RNA-based delivery is highly efficient. However, in contrast to this conventional mRNA approach, alphavirus RNA replicons provide the additional advantage of RNA self-amplification leading to superior expression levels. Moreover, replacement of the SP6 RNA polymerase promoter by a CMV promoter, DNA replicon vectors for GoI expression (Figure 2C) have been engineered for the transfection of cell lines and in vivo administration [14]. The use of DNA replicons eliminates any risk of the production of new virus particles but relies on the less efficient delivery of DNA compared to viral vectors. Moreover, DNA molecules must be delivered to the nucleus for the in vivo transcription of RNA (Figure 2C).

## 3. Alphavirus-Based Immunotherapy for Cancer

In the context of cancers, alphaviruses have been frequently used for prophylactic and therapeutic applications. Immunization with alphavirus vectors overexpressing tumor-associated antigens (TAAs) has been a common approach for cancer vaccine development. This approach has been used to provide both prevention against tumor challenges and tumor regression and eradication. Moreover, overexpression of cytotoxic and antitumor genes has been evaluated for cancer therapy. The delivery of immunostimulatory genes from alphavirus vectors has served the means of cancer immunotherapy. Moreover, alphaviruses induce apoptosis through activation of caspases in infected cells [15], which has resulted in tumor regression after administration of alphaviruses carrying no therapeutic genes and has allowed the use of vectors with reporter genes to verify and localize expression in animal tumor models. Finally, engineered or naturally occurring oncolytic alphaviruses have demonstrated tumor cell-specific killing in animal models [16]. Examples of cancer vaccinations, cancer therapy and immunotherapy are given below and summarized in Table 1.

### 3.1. Reporter Genes

Due to the apoptosis induced by alphaviruses, therapeutic efficacy can be achieved by the delivery of alphavirus vectors carrying no foreign genes [15]. However, the introduction of reporter genes has substantially facilitated the monitoring of transgene delivery and expression. For example, SFV particles expressing the enhanced green fluorescent protein (EGFP) showed the efficient killing of human H358a nonsmall cell lung cancer (NSCLC) cells and inhibited the growth of H358a spheroids [17]. Complete tumor regression was achieved in three out of seven nu/nu mice carrying H358a xenografts after injection of SFV-EGFP particles. In another approach, SIN-LacZ particles were intravenously administered to mice implanted with CT26/CL25 colon tumors, which provided complete tumor remission and long-term survival [18]. Moreover, a single intramuscular injection of 0.1 μg SFV-LacZ RNA replicon induced strong antigen-specific immune responses in mice and provided protection against tumor challenges with CT26 tumor cells [19]. The regression of pre-existing tumors was also observed resulting in prolonged survival of mice.

### 3.2. Tumor-Associated Antigens

Overexpression of TAAs from alphavirus vectors has been a common strategy for inducing immune responses for the development of cancer vaccines. For example, VEE particles expressing the human papilloma virus type 16 (HPV-16) envelope E7 protein elicited robust CD8^+^ T cell responses and protected immunized C57BL/6 mice from HPV-16 challenges [20]. Moreover, immunization of C57BL/6 mice with the SFVenh vector expressing the HPV E6-E7 protein resulted in complete eradication of existing tumors and induced CTL responses lasting for 340 days [21]. In another approach, fusion of helper T cell epitopes and an endoplasmic reticulum (ER) targeting signal to the HPV E6 and E7 proteins resulted in the SFV-sHELP-E7SH vector, which generated tumor regression and protection against tumor challenges in C57BL/6JOlaHSd mice [22]. In the context of DNA replicons, SFV-HPV E6-E7 DNA plasmids were intradermally administered to C57BL/6 mice followed by electroporation [23]. Immunization with an SFV-HPV E6-E7 DNA dose of 0.05 μg, which is 200-fold lower than that used for conventional DNA plasmids, efficiently prevented tumor growth, resulting in 85% of mice becoming tumor-free. SFVenh-HPV E6-E7 particles have been evaluated in 12 patients with a history of cervical intraepithelial neoplasia in a phase I clinical trial [24]. Immunization with 5 × 10^5^, 5 × 10^6^, 5 × 10^7^ or 2.5 × 10^8^ particles showed a good safety profile and elicited HPV-specific immune responses in all vaccinated patients.

The vascular endothelial growth factor receptor-2 (VEGFR-2) has also been expressed from SFV vectors [25]. Immunization of mice with SFV-VEFR-2 particles significantly inhibited tumor growth and metastatic spread. Interestingly, co-immunization with SFV particles expressing interleukin-12 (IL-12) reduced antitumor activity. In contrast, co-immunization of SFV-VEGFR-2 and SFV-IL-4 particles was superior to immunization with only SFV-VEGFR-2 particles and prolonged the survival of mice [25]. The carcinoembryonic antigen (CEA) has been expressed from VEE vectors in clinical trials [26,27]. In this context, VEE-CEA particles have been subjected to a phase I study in patients with stage III and IV colorectal cancer [26]. Antigen-specific immune responses were generated, providing long-term survival among both stage III and IV patients. In another phase I study, VEE-CEA particles showed efficient transduction of dendritic cells (DCs) and was repeatedly administered to patients with metastatic pancreatic cancer [27]. VEE-CEA elicited clinically relevant CEA-specific T cell and antibody responses. Moreover, VEE-CEA generated cytotoxicity against tumor cells from human colorectal metastases and resulted in prolonged survival of pancreatic cancer patients. Recombinant VEE particles expressing the tyrosine-related protein-2 (TRP-2) have been compared to a plasmid DNA-based vaccine in a mouse B16F10 melanoma model [28]. VEE-TRP-2 particles induced immune responses and provided tumor protection in mice. Moreover, a heterologous DNA prime immunization followed by a VEE-TRP-2 booster showed superior immunogenicity and protection compared to the administration of plasmid DNA alone. In another study, VEE-TRP-2 particles elicited humoral immune responses, antitumor activity and prolonged survival in immunized mice implanted with B16 tumors [29]. Superior tumor regression was obtained by combining VEE-TRP-2 administration with antagonist anti-CTL antigen-4 (CTLA-4) or agonist anti-glucocorticoid-induced tumor necrosis factor receptor (GITR) monoclonal antibodies (mAbs) [30]. Co-administration of VEE-TRP-2 particles with CTLA-4 mAbs and GITR mAbs resulted in tumor regression in 50% and 90% of mice, respectively [30]. In another approach, the SFV DNA replicon expressing VEGFR-2 and IL-12 was co-administered with another SFV DNA replicon expressing survivin and β-hCG antigens to mice with B16 melanoma xenografts [31]. The combination therapy provided superior antitumor activity compared to administration of either SFV DNA replicon alone.

In the case of ovarian cancer, SFV particles expressing ovalbumin (OVA) have been subjected to a prime-boost immunization strategy with vaccinia virus (VV-OVA) in C57BL/6 mice [32]. This approach elicited OVA-specific CD8^+^ T cell responses and enhanced antitumor activity in mice with murine ovarian surface epithelial carcinoma (MOSEC). VEE particles have also been subjected to immunization in mouse prostate tumor models. For example, expression of the prostate-specific membrane antigen (PSMA) in BALB/c and C57BL/6 mice elicited strong PSMA-specific immune responses [33]. Robust T and B-cell responses were obtained after a single immunization, and the immune responses were enhanced after repeated VEE-PSMA administration. Moreover, VEE-PSMA particles have been evaluated in castration-resistant metastatic prostate cancer (CRPC) patients in a phase I study [34]. Although immunization demonstrated good safety and tolerability, the immune responses were disappointingly weak. In another approach, the prostate-specific antigen (PSA) was expressed from a VEE vector and evaluated for immunogenicity and tumor growth inhibition in mice [35]. The outcome was a strong stimulation of the production of IgG2a/b anti-PSA antibodies and a significant delay in tumor growth. Moreover, VEE-PSA immunizations efficiently overcame immune tolerance to PSA and mediated rapid clearance of PSA-expressing tumor cells. VEE particles have also been utilized for the expression of the mouse six-transmembrane epithelial antigen of the prostate (mSTEAP) [36]. After immunization with VEE-mSTEAP, a booster immunization with gold-coated conventional pcDNA-3-mSTEAP plasmids using gene gun technology, elicited mSTEAP-specific immune responses, significantly prolonged survival and protected mice against tumor challenges. In another study, VEE vectors have been used for the expression of the prostate stem cell antigen (PSCA) [37]. Immunization studies in transgenic adenocarcinoma of the prostate (TRAMP) mice resulted in the survival of 90% of mice for at least one year.

### 3.3. Cytotoxic and Antitumor Genes

As endostatin specifically inhibits the endothelial proliferation and potently inhibits angiogenesis and tumor growth, it has been used for tumor therapy. SFV particles expressing endostatin were compared to the administration of retrovirus–expressed endostatin and SFV-LacZ particles in a mouse B16 brain tumor model [38]. Clearly, SFV–endostatin treatment was superior to retrovirus- or SFV-LacZ-based therapy in relation to tumor growth inhibition and reduction in intratumoral vascularization.

In the context of breast cancer, the human epidermal growth factor receptor (HER2/neu), a proto-oncogene, has been an attractive target for cancer therapy. For example, HER2/neu expression from a SIN DNA replicon has been compared to adenovirus-based HER2/neu expression [39]. Both SIN DNA replicons and adenovirus particles generated significant inhibition of tumor growth in mice when the immunization took place before the tumor challenge. In contrast, the efficacy was low when the immunization took place two days after the tumor challenge. The combination of the prime immunization of SIN DNA replicons followed by adenovirus particle administration resulted in significantly prolonged survival. It has also been demonstrated that 80% less SIN-HER2/neu replicon DNA was needed to elicit strong antibody responses compared to conventional plasmid DNA [40]. In another study, VEE particles have been engineered to express the extracellular domain (ECD) and transmembrane (TM) domains of HER2 [41]. VEE-HER2 ECD/TM particles injected into mammary tissue or intravenously administered prevented or inhibited the growth of mouse breast cancer cells expressing HER2/neu. The immunization elicited HER2/neu-specific CD8^+^ T lymphocytes and serum IgG and resulted in complete prevention of tumor formation in mice. VEE-HER2 particles have also been subjected to a phase I clinical trial in stage IV HER2 overexpressing breast cancer patients [42]. The treatment showed good safety and tolerability. Partial response (PR) was detected in one patient, and stable disease (SD) was achieved in two other patients.

### 3.4. Immunostimulatory Genes

Immunotherapy has become an attractive part of cancer treatment and immunostimulatory proteins have played a central role. Due to limitations in the injection of purified recombinant proteins, administration of viral vectors expressing immunostimulatory genes has been shown to be a promising strategy. For example, SFV particles expressing IL-18 were transduced into DCs and co-administered with the IL-12 protein systemically [43]. Enhanced Th1 responses in tumor-specific CD4^+^ and CD8^+^ T cells and natural killer cells in mice were superior after treatment with SFV-IL-18 and recombinant IL-12 than with IL-12 alone. Likewise, the antitumor activity and protective immunity were superior after combination therapy. SFV-IL-12 particles have been administered via an implanted canula in a syngeneic RG2 rat glioma model [44]. Administration of 5 × 10^7^ and 5 × 10^8^ SFV-IL-12 particles resulted in 70% and 87% reduction in tumor volume, respectively. While the lower dose significantly extended the time of survival, the higher dose could potentially induce vector-related lethal pathology. SIN DNA replicons expressing human gp100 and mouse IL-18 were intramuscularly injected into mice implanted with B16-gp100 brain tumors [45]. Combination therapy with SIN-gp100 and SIN-IL-18 DNA replicons provided superior therapeutic and prophylactic activity against brain tumors and prolonged the survival of mice. Another combination therapy involved administration of SFV-IL-12 particles and the *Salmonella typhimurium* aroC strain (LV101), which resulted in complete inhibition of lung metastasis formation and extended survival in 90% of mice with implanted 4T1 mammary tumors [46]. The combination approach provided a synergistic antitumor effect substantially superior to treatment with either SFV-IL-12 particles or LV101 alone. Moreover, single immunization of SFVenh-IL-12 particles in mice implanted with MC38 colon tumor xenografts resulted in complete tumor regression and long-term tumor-free survival [47]. Repeated intratumoral injections improved antitumor activity and were superior in the elimination of tumors when compared to a first-generation adenovirus vector. In another approach, SFV-IL-12 particles were coadministered with the immune checkpoint inhibitor, anti-PD1 monoclonal antibody, which resulted in a superior synergistic effect in the MC38 mouse colon tumor model compared to SFV-IL-12 alone [48]. The synergistic effect was further confirmed in a B16-OVA melanoma mouse model [48]. In addition, VEE particles have been utilized for the expression of IL-12 [49]. Immunization of C57BL/6 mice implanted with MC38-CEA-2 tumors with VEE-IL-12 showed superior tumor growth inhibition compared to recombinant IL-12 administration [49]. However, co-administration of VEE-IL-12 and VEE-CEA elicited superior immune responses and antitumor activity and prolonged the survival of immunized mice [49]. Liposome-encapsulated SFV particles expressing IL-12 (LSFV-IL-12) have been subjected to 18 patients with stage III or IV metastasizing melanoma and kidney carcinoma in a phase I study [50]. Intravenous administration of LSFV-IL-12 particles did not cause any major toxicity, only mild inflammatory reactions. Ten-fold increase in the IL-12 concentrations were detected in peripheral blood, which lasted for 3–4 days. Moreover, repeated LSFV-IL-12 administration showed good safety and tolerability. SIN particles expressing IL-12 have been co-administered with the CPT-11 topoisomerase inhibitor irinotecan to SCID mice with highly aggressive ES2 ovarian tumors [51]. Long-term survival was established in 35% of immunized mice, which was not the case for the treatment with either SIN-IL-12 or irinotecan alone.

### 3.5. Oncolytic Viruses

Oncolytic viruses have shown efficacy in cancer therapy as demonstrated by the approval of the oncolytic herpesvirus-based drug talimogene laherparepvec (T-VEC) for melanoma treatment [60]. Oncolytic alphaviruses have also shown great promise in cancer treatment. For example, the replication-competent SFV(A774nsP) vector expressing EGFP (SFV VA-EGFP) has been evaluated in a subcutaneous orthotopic glioblastoma mouse model [52]. A single intravenous injection of SFV-VA-EGFP completely inhibited the stable expression of firefly luciferase (Luc) in orthotopic U87Fluc tumors. Moreover, long-term survival was established in 16 out of 17 mice. In another study, the SFV VA-EGFP vector efficiently eradicated subcutaneous and orthotopic human prostate tumors in BALB/c mice [53]. The SFV VA-EGFP has also been compared to a conditionally replicating adenovirus Ad-Delta24TK-GFP demonstrating superior survival of SFV VA-EGFP immunized mice with A549 lung adenocarcinoma implants [8]. However, systemic delivery of SFV or Ad vectors did not elicit any significant immune response.

The naturally oncolytic M1 alphavirus can selectively infect and kill zinc finger antiviral protein (ZAP)-deficient tumor cells without causing damage to normal cells [61]. The key stages of infection and replication of M1 have been visualized using transmission electron microscopy (TEM) in the HepG2 liver cancer cell line [62]. M1 induced typical apoptotic events such as vacuolization of cancer cells and nuclear marginalization. Furthermore, engineering of an M1-GFP vector allowed visualization and quantification of M1 in vitro and in vivo [54]. M1-GFP showed stable reporter gene expression for at least 10 generations in the colorectal SW620 cell line and tumor targeting in BALB/c mice with implanted Hep3B liver tumors [54]. In another study conducted in rats, cynomolgus macaques and mice with implanted tumors showed gradual elimination of M1 from normal tissue, whereas M1 replication was prominent in tumor tissue [55]. Moreover, M1 infiltrated the blood-brain-barrier and replicated in malignant glioma cells, leading to their efficient killing. The oncolytic activity of M1 was demonstrated in an orthotopic mouse bladder cancer model, which was further improved by knocking down the coiled-coiled domain containing 6 (CCDC6) gene in M1-infected mice [63]. In the context of bladder cancer, M1 significantly decreased cell viability in eight bladder cancer cell lines but not in normal bladder cells [56]. Moreover, evaluation of M1-GFP demonstrated replication in T24 and UM-UC-3 bladder cell lines and in primary patient-derived bladder cells but not in normal cells. Tail vein injections of orthotopic muscle-invasive bladder cancer (MIBC) mice resulted in significant inhibition of tumor growth and prolonged survival [56]. The M1 treatment provided a stronger antitumor effect than cisplatin treatment. M1 has also been subjected to experimental treatment of the highly aggressive triple-negative breast cancer (TNBC) [7]. In vitro, the oncolytic activity was enhanced 100-fold by co-administration of doxorubicin. Moreover, the tumor growth significantly slowed down in vivo. To further improve the oncolytic effect of the M1 alphavirus, delivery has been combined with irreversible electroporation (IRE), also called the nanoknife [57]. The apoptotic effect of IRE and the oncolytic activity of M1 provided a synergistic effect in pancreatic cancer cells. In vivo, the IRE-M1 combination therapy enhanced the inhibition of tumor proliferation and prolonged the survival of immunocompetent mice with orthotopic pancreatic tumors [57]. In a strategy to attenuate the immunogenicity of M1, the virus was encapsulated in liposomes (M-LPO) [58]. It was demonstrated that M-LPO inhibited the growth of colorectal LoVo and liver Hep 3B cancer cells, whereas naked M1, only liposomes or a mixture of liposomes and M1, did not [58]. Intravenous injections into mice reduced the production of M1-specific neutralizing antibodies and may reduce the intrinsic viral immunogenicity, providing improved anticancer therapy.

The replication-competent SIN AR339 strain has been shown to induce cytopathogenicity and apoptosis in cancer cells but not in normal cells in vitro [59]. In nude mice, SIN AR339 caused significant regression of established cervical tumors after a single intraperitoneal or intravenous injection [59]. Moreover, SIN AR339 suppressed the formation of ascites in a metastasis model of ovarian cancer in nude mice [59].

## 4. Conclusions

Alphaviruses have been utilized for efficient high-level expression of TAAs, cytotoxic and antitumor genes and immunostimulatory genes with the goal of treating various cancers. Moreover, oncolytic alphaviruses specifically infecting and killing tumor cells without affecting normal cells have also presented an attractive approach. As presented above and summarized in Table 1, numerous examples in animal tumor models have demonstrated the inhibition of tumor growth, eradication of tumors, cure of disease and protection against tumor challenges.

In the context of clinical trials, although safety and tolerability have been good and specific immune responses have been recorded, therapeutic and protective efficacy has been relatively modest indicating that genuine challenges remain. For example, the transition from proof of concept studies to clinical applications in human subjects has not been as smooth as originally anticipated. Suitable TAA antigens have not been identified for all types of cancers. Moreover, although clinical benefits have been established in rodents, prophylactic and therapeutic efficacy in humans has not been obtained due to inefficient delivery or use of suboptimal doses. Another issue relates to the discrepancy between existing animal models and clinical settings in humans. Rodent tumor models, in particular, rely on tumor cells artificially grafted on various locations in the target animals. In contrast, human tumors spontaneously develop setting a totally different environment and conditions for therapeutic interventions. For this reason, canine cancer models based on the spontaneous appearance of tumors in dogs might be a better approach for preclinical evaluations. It might also provide the means for developing novel veterinary immunotherapy and anticancer therapy, although it should not be understood as a potential shortcut to clinical development in humans. It is also essential to acknowledge that alphavirus-based approaches might not be suitable for all types of cancers, although a large variety of cancers have been targeted (Table 1). For this reason, it might be advantageous to focus future activities on certain types of cancers. For example, brain tumors and pancreatic cancers can be considered as attractive targets due to the generally poor prognosis and lack of efficient alternative approaches. The positive initial findings from combination therapy with other viral vectors and/or existing cancer drugs and immune checkpoint inhibitors present interesting potential opportunities for alphavirus-based gene therapy.

The flexibility of application of alphaviruses as recombinant replication-deficient and -competent particles, naked or encapsulated RNA replicons and plasmid-based DNA replicons should also improve the chance of success. The induction of apoptosis by alphaviruses and the transient nature of expression are features which can also contribute to therapeutic efficacy and safety. Certainly, the biggest asset of alphaviruses for the development of vaccines and therapeutics is the RNA self-replication, which enhances the immune responses and allows the use of 100-1000-fold lower doses of RNA or DNA compared to conventional synthetic mRNA or plasmid DNA.

## Figures and Tables

**Figure 1 biomedicines-10-02263-f001:**
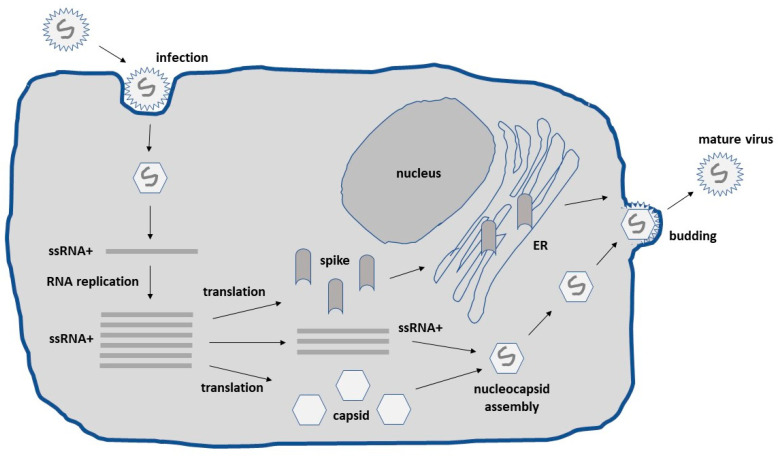
**Schematic presentation of the lifecycle of alphaviruses.** Alphaviruses infect host cells by endocytosis through endosomal fusion with the plasma membrane. The positive sense ssRNA is released into the cytoplasm for translation of viral proteins and RNA replication. Full-length ssRNA genomes are packaged into nucleocapsids. The alphavirus envelope proteins are transported to the plasma membrane through the endoplasmic reticulum and Golgi. The nucleocapsids are encircled by the envelope proteins at the plasma membrane and released by budding. ER, endoplasmic reticulum.

**Figure 2 biomedicines-10-02263-f002:**
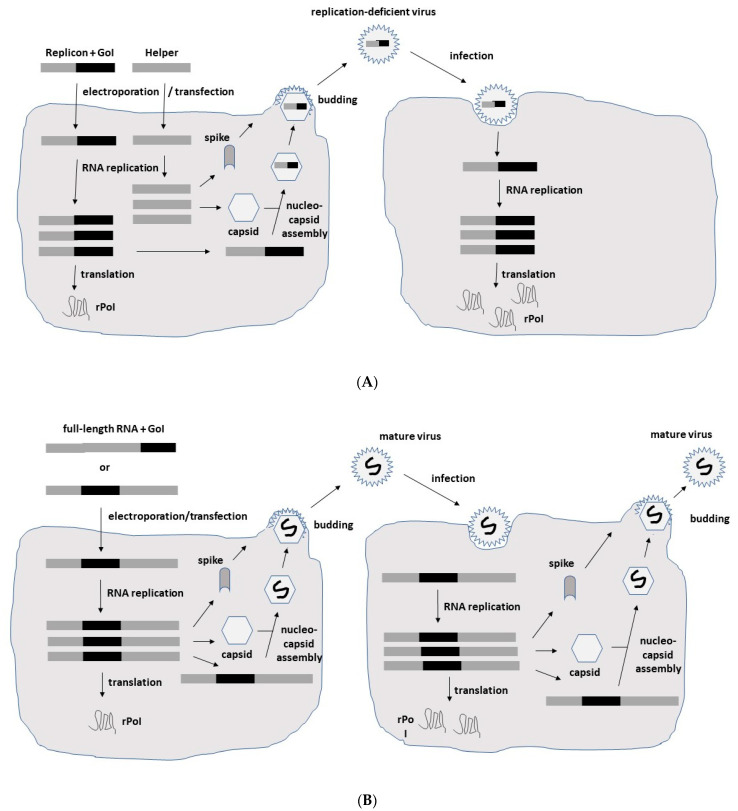
**Schematic presentation of SFV expression systems.** (**A**) **Replication-deficient recombinant particles.** In vitro transcribed RNA molecules from the SFV expression vector carrying the nonstructural protein (nsP) genes, replicase genes (replicon) and the gene of interest (GoI) and the structural protein genes (capsid, 6K, envelope E1, E2 and E3) from the helper vector are electroporated or transfected into BHK-21 cells. After RNA replication, only the RNA from the expression vector containing the packaging signal is packaged into nucleocapsids and transported to the plasma membrane, where budding of mature viral particles takes place. Although the generated particles are capable of infecting new host cells, no viral progeny is produced due to the absence of the structural protein genes. However, high-level expression of the recombinant protein of interest (rPoI) takes place (**B**) **Replication-competent recombinant particles.** The in vitro transcribed full-length RNA genome with the GoI introduced either downstream of the nsP genes or the structural protein genes is electroporated or transfected into host cells for production of replication-competent viral particles and rPoI expression. (**C**) **DNA replicon vectors.** The replacement of the SP6 RNA polymerase promoter by a CMV promoter upstream of the nsP genes allows for direct transfection of host cells for rPoI expression. DNA replicons in the form of DNA plasmids are transfected into host cells, and DNA replicons are delivered to the nucleus. Transcribed ssRNA molecules of positive polarity are delivered to the cytoplasm for RNA replication and expression of the rPoI.

**Table 1 biomedicines-10-02263-t001:** Examples of alphavirus-based vaccines against cancer.

Cancer	Vector	Finding	Ref
**Reporter Genes**			
Lung	SFV-EGFP	Tumor regression in mice	[17]
Colon	SIN-LacZ	Complete tumor remission	[18]
	SFV-LacZ RNA	Tumor regression, protection	[19]
**TAAs**			
Cervical	VEE-HPV-16 E7	Protection against tumor challenges in mice	[20]
	SFVenh-HPV E6-E7	Tumor eradication, long-lasting CTL in mice	[21]
	SFV-sHELP-E7SH	Tumor regression, protection in mice	[22]
	SFV-HPV E6-E7 DNA	85% of immunized mice tumor-free	[23]
	SFVenh-HPV E6-E7	Phase I: Immunogenicity in all patients	[24]
Colon	SFV-VEGFR-2	Inhibition of tumor growth, metastatic spread	[25]
	SFV-VEGR-2 + SFV-IL-4	Prolonged survival after coadministration	[25]
	VEE-CEA	Phase I: Ag-specific response, long-term survival	[26]
Pancreatic	VEE-CEA	Phase I: Prolonged survival	[27]
Melanoma	VEE-TRP-2 + DNA	Superior to plasmid DNA vaccine in mice	[28]
	VEE-TRP-2	Humoral immune responses, protection in mice	[29]
	VEE-TRP-2 + CTLA-4 mAbs	Tumor regression in 50% of mice	[30]
	VEE-TRP-2 + GITR mAbs	Tumor regression in 90% of mice	[30]
	SFV-VEGFR-2/IL-12 DNA	Synergistic antitumor activity from combination of	[31]
	+ SFV-Survivin/β-hCG DNA	DNA replicons	
Ovarian	SFV-OVA + VV-OVA	Immune responses, enhanced antitumor activity	[32]
Prostate	VEE-PSMA	Th1-biased immune responses	[33]
	VEE-PSMA	Phase I: Good safety, weak immunogenicity	[34]
	VEE-PSA	PSA-specific Abs, delay in tumor growth	[35]
	VEE-mSTEAP + pcDNA	Prolonged survival, tumor challenge protection	[36]
	VEE-PSCA	Long-term survival of mice	[37]
**Cytotoxic and Antitumor Genes**			
Glioblastoma	SFV–Endostatin	Tumor growth inhibition, reduced vascularization	[38]
Breast	SIN-HER2/neu DNA	Significant tumor growth inhibition, protection	[39]
	SIN-HER2/neu DNA	80% less DNA needed compared to plasmid DNA	[40]
	VEE-HER2/neu ECD/TM	Complete prevention of tumors in mice	[41]
	VEE-HER2/neu ECD/TM	Safe, PR in 1 patient, SD in 2 patients	[42]
**Immunostimulation**			
Glioblastoma	SFV-IL-18 + rec IL-12	Superior therapeutic effect of combination	[43]
Glioma	SFV-IL-12	70–97% tumor volume reduction in rats	[44]
Brain	SIN-gp100 + SIN-IL-12 DNA	Superior antitumor activity, prolonged survival	[45]
Breast	SFV-IL-12 + LV101	Superior antitumor activity of combination	[46]
Colon	SFVenh-IL-12	Complete tumor regression, long-term survival	[47]
	SFV-IL-12 + anti-PD1	Superior combination therapy in mice	[48]
	VEE-IL-12 + VEE-CEA	Superior combination therapy in mice	[49]
Melanoma	SFV-IL-12 + anti-PD1	Superior combination therapy in mice	[48]
	LSFV-IL-12	Phase I: Good safety and tolerability	[50]
Ovarian	SIN-IL12 + Irinotecan	Long-term survival in 35% of mice	[51]
**Oncolytic Viruses**			
Glioblastoma	SFV-VA-EGFP	Long-term survival in 16 out of 17 mice	[52]
Prostate	SFV-VA-EGFP	Complete tumor eradication in mice	[53]
Lung	SFV-VA-EGFP	Long-term survival in mice	[8]
Liver	M1	Liver tumor targeting in mice	[54]
Glioma	M1	Replication in tumors	[55]
Bladder	M1	Tumor growth inhibition, prolonged survival	[56]
Breast	M1 + Doxorubicin	Reduced tumor growth in mice	[7]
Pancreatic	M1 + IRE	Tumor growth inhibition, prolonged survival	[57]
Cervical	SIN AR339	Regression of established tumors in mice	[58]
Ovarian	SIN AR399	Ascites formation in metastasis mouse model	[59]

Abs, antibodies; anti-PD1, immune checkpoint inhibitor; CEA, carcinoembryonic antigen; CTLA-4, CTL antigen-4; GITR, glucocorticoid-induced tumor necrosis factor; HPV, human papilloma virus; IRE, irreversible electroporation; LV101, *Salmonella typhimurium* AroC strain; mAbs, monoclonal antibodies; mSTEAP, mouse six-transmembrane epithelial antigen of the prostate; OVA. Ovalbumin; PR, partial response; PSA, prostate-specific antigen; PSMA, prostate-specific membrane antigen; rec IL-12, recombinant IL-12; SD, stable disease; SFV, Semliki Forest virus; SIN, Sindbis virus; TRP-2, tyrosine-related protein-2; VEE, Venezuelan equine encephalitis virus; VEGFR-2, vascular endothelial growth factor receptor-2; VV, vaccinia virus.

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
