# Peer review of "Alphaviruses in Immunotherapy and Anticancer Therapy"

_biomedicines, 2022, doi:10.3390/biomedicines10092263_

Round 1

Reviewer 1 Report

This review by Lundstrom is complete and nicely summarizes the research on alphaviruses used for cancer therapy. The author has been researching this area for more than 20 years and has a wealth of information included in the review.

Major comments:

1. In the past 3 years the M1 virus has been studied and the author should include more of these papers when discussing oncolytic alphaviruses. It felt the incorporation of the M1 virus was an afterthought or just tacked on the end and not given the full thought the rest of the review has.

2. After reading the review, the reader thinks Alphaviruses are fantastic and should be used for all cancer therapies and eventually for other diseases. This has not happened and clearly for scientific reasons. It would benefit the reivew to have a section on "Challenges of using Alphaviruses" and list the negative data, the experimental issues, etc. This will also focus the reader on areas where efforts should be made to move alphaviruses to clinical trials.

Author Response

This review by Lundstrom is complete and nicely summarizes the research on alphaviruses used for cancer therapy. The author has been researching this area for more than 20 years and has a wealth of information included in the review.

Major comments:

  1. In the past 3 years the M1 virus has been studied and the author should include more of these papers when discussing oncolytic alphaviruses. It felt the incorporation of the M1 virus was an afterthought or just tacked on the end and not given the full thought the rest of the review has.

Response: I respectfully disagree with the comment that the presentation of M1 virus was an afterthought. In the section on oncolytic alphaviruses, the engineered oncolytic SFV vectors are presented first followed by M1 and ending with the naturally oncolytic SIN AR339. However, 6 published papers on M1 have now been added, described in the text, and cited in References.

  1. After reading the review, the reader thinks Alphaviruses are fantastic and should be used for all cancer therapies and eventually for other diseases. This has not happened and clearly for scientific reasons. It would benefit the reivew to have a section on "Challenges of using Alphaviruses" and list the negative data, the experimental issues, etc. This will also focus the reader on areas where efforts should be made to move alphaviruses to clinical trials.

Response: A section on challenges of using alphaviruses has been added to the Conclusions section.

Reviewer 2 Report

Alpha viruses can replicate in the cytoplasm of the infected cell due to the replicase genes. Modified alphaviruses (replicon and heterologous gene of interest, or whole RNA and heterologous gene of interest) serve as vectors and their transfection expresses the gene of interest with/without release of complete virions. The complex technique enables the expression of tumor-related genes (and a cytotoxic immune response to them), sometimes in promising combination with targeted immunotherapy or immunomodulation. The article is informative, the topic is modern and exciting, I recommend it for publication.

Minor comments:
In my opinion, the strength of the manuscript is its informativeness about the pre-clinical/to-date clinical application of alphaviruses as vectors for gene therapy. The "weakness" is in the concise and very professional presentation of facts, which is difficult to follow for a person who is not closely related to this field.

In the discussion, explain the possibilities of clinical application of gene therapy using alphavirus as a vector in the near future.

Figure 2 C. The non-expert reader needs a simpler and broader explanation to understand it.

Author Response

Alpha viruses can replicate in the cytoplasm of the infected cell due to the replicase genes. Modified alphaviruses (replicon and heterologous gene of interest, or whole RNA and heterologous gene of interest) serve as vectors and their transfection expresses the gene of interest with/without release of complete virions. The complex technique enables the expression of tumor-related genes (and a cytotoxic immune response to them), sometimes in promising combination with targeted immunotherapy or immunomodulation. The article is informative, the topic is modern and exciting, I recommend it for publication.

Minor comments:
In my opinion, the strength of the manuscript is its informativeness about the pre-clinical/to-date clinical application of alphaviruses as vectors for gene therapy. The "weakness" is in the concise and very professional presentation of facts, which is difficult to follow for a person who is not closely related to this field.

Response: I acknowledge the point the reviewer is making. However, Biomedicines is a journal aimed at professionals in the field and not a popularized scientific magazine or textbook reaching out to the wider scientific community or the general population.

In the discussion, explain the possibilities of clinical application of gene therapy using alphavirus as a vector in the near future.

Response: Text has been added to the Conclusions section.

Figure 2 C. The non-expert reader needs a simpler and broader explanation to understand it.

Response: Text has been added to Fig. 2C to provide additional information on how the DNA replicons work.  

Reviewer 3 Report

The manuscript by Kenneth Lundstrom describes an interesting review on the use of alphaviruses in immunotherapy of cancer diseases. The author describes the use of alphavirus vectors expressing tumor-associated antigens (TAAs) , cytotoxic and anti-tumor genes or immunostimulatory genes in anti-cancer therapy. In addition, the author describes the use of engineered or naturally occuring alphaviruses in cancer in vitro and in vivo models. This manuscript is well-written, however it is very similar to author’s recently published paper in Frontiers in Molecular Biosciences (2022) titled „Alphaviruses in cancer therapy”. Moreover, the present article also describes the use of alphaviruses in anticancer therapy, and the title of this manuscript should be adequatly changed, as it does not describe the use of these viruses in broadly understood immunotherapy. Minor issues: Line 196: „ovarian” instead of „ovatian” Line 308: „overcame” instead of „overcoame” Line 262: „Salmonella Typhimurium”, where Salmonella should be in italics Lines 308-309: additional paragraph      

Author Response

The manuscript by Kenneth Lundstrom describes an interesting review on the use of alphaviruses in immunotherapy of cancer diseases. The author describes the use of alphavirus vectors expressing tumor-associated antigens (TAAs) , cytotoxic and anti-tumor genes or immunostimulatory genes in anti-cancer therapy. In addition, the author describes the use of engineered or naturally occurring alphaviruses in cancer in vitro and in vivo models. This manuscript is well-written, however it is very similar to author’s recently published paper in Frontiers in Molecular Biosciences (2022) titled „Alphaviruses in cancer therapy”. Moreover, the present article also describes the use of alphaviruses in anticancer therapy, and the title of this manuscript should be adequately changed, as it does not describe the use of these viruses in broadly understood immunotherapy.

Response: “Anticancer therapy” has been added to the title.

Minor issues: Line 196: „ovarian” instead of „ovatian” Line 308: „overcame” instead of „overcoame” Line 262: „Salmonella Typhimurium”, where Salmonella should be in italics Lines 308-309: additional paragraph

Response: Thank you for spotting these typing errors. They have been corrected including other errors spotted.